# Exploring the Biological Properties of Zn(II) *Bis*thiosemicarbazone Helicates

**DOI:** 10.3390/ijms24032246

**Published:** 2023-01-23

**Authors:** Sandra Fernández-Fariña, Isabel Velo-Heleno, Rocío Carballido, Miguel Martínez-Calvo, Ramiro Barcia, Òscar Palacios, Mercè Capdevila, Ana M. González-Noya, Rosa Pedrido

**Affiliations:** 1Departamento de Química Inorgánica, Facultade de Química, Campus Vida, Universidade de Santiago de Compostela, 15782 Santiago de Compostela, Spain; 2Departamento de Bioquímica y Biología Molecular, Facultade de Veterinaria, Campus Terra, Universidade de Santiago de Compostela, 27002 Lugo, Spain; 3Departament de Química, Universitat Autònoma de Barcelona, 08193 Cerdanyola del Vallès, Spain

**Keywords:** bisthiosemicarbazone ligands, zinc, helicates, biological activity

## Abstract

The design of artificial helicoidal molecules derived from metal ions with biological properties is one of the objectives within metallosupramolecular chemistry. Herein, we report three zinc helicates derived from a family of *bis*thiosemicarbazone ligands with different terminal groups, Zn_2_(L^Me^)_2_∙2H_2_O **1**, Zn_2_(L^Ph^)_2_∙2H_2_O **2** and Zn_2_(L^PhNO2^)_2_
**3**, obtained by an electrochemical methodology. These helicates have been fully characterized by different techniques, including X-ray diffraction. Biological studies of the zinc(II) helicates such as toxicity assays with erythrocytes and interaction studies with proteins and oligonucleotides were performed, demonstrating in all cases low toxicity and an absence of covalent interaction with the proteins and oligonucleotides. The in vitro cytotoxicity of the helicates was tested against MCF-7 (human breast carcinoma), A2780 (human ovarian carcinoma cells), NCI-H460 (human lung carcinoma cells) and MRC-5 (normal human lung fibroblasts), comparing the IC_50_ values with cisplatin. We will try to demonstrate if the terminal substituent of the ligand precursor exerts any effect in toxicity or in the antitumor activity of the zinc helicates.

## 1. Introduction

Thiosemicarbazones are well-known organic motifs due to their versatility on coordination [1,2,3] and their broad therapeutic activity [4,5,6,7,8,9,10]. In this sense, thiosemicarbazone ligands and many of their complexes have been tested as antitumor agents, some compounds reaching phases I–III of clinical trials [11,12]. They have also been tested as antibacterial [13,14,15] and antifungal [16,17] therapeutics, among many other pathologies. All these studies make clear the need to know in depth the biochemical targets of thiosemicarbazone complexes, or the transformations that they may experience in physiological media.

On the other hand, in the last decades there has been notable progress in the development of selective pathways driving to particular supramolecular architectures with biomedical applications [18,19,20]. Among them, helicates can be considered very promising candidates that can be precisely designed to exhibit inherent bioactivity. The term “helicate” refers to a particular class of self-assembled metallosupramolecular compounds, in which conveniently designed ligands wrap around two or more metal ions and form a double stranded helix [21,22,23,24,25].

In the last few years, our group has demonstrated that pentadentate *bis*thiosemicarbazones can act as useful building blocks for generating novel supramolecular motifs, such as helicates, cluster helicates or mesocates [26,27]. Moreover, we have demonstrated that the nuclearity and the ligand arrangement exhibited by pentadentate thiosemicarbazone complexes depend on the metal size and the deprotonation degree of the ligand [28]. In addition, the synthetic procedure employed can determine the shape and nuclearity of the final supramolecular arrangement. More in particular, Zn(II) ions in combination with dianionic *bis*thiosemicarbazones featuring a pyridine spacer give rise to dinuclear *bis*helicoidal Zn(II) neutral complexes displaying different internal coordination modes for the two zinc ions that can be described as [6+6], [6+4], [5+5] and [4+4] in the solid state [28,29,30,31,32].

With this in mind, in this work, we have combined thiosemicarbazone skeletons and zinc ions in the search for helical arrangements that could be non-toxic candidates for new therapeutic agents. Thus, in this work we report three new zinc helicates derived from the *bis*thiosemicarbazone ligands *bis*(N(4)-R-thiosemicarbazone)-2,6-formylpyridine (R = Me, Ph, PhNO_2_) obtained by means of an electrochemical procedure, together with their crystal structures. Furthermore, we have explored the biological potential of the zinc(II) helicates by performing toxicity assays and studying their interaction with biological targets such as proteins and oligonucleotides. Moreover, the in vitro cytotoxicity was tested against three cancer cell lines: MCF-7 (human breast carcinoma), A2780 (human ovarian carcinoma cells) and NCI-H460 (human lung carcinoma cells), and with the healthy cell line MRC-5 (normal human lung fibroblasts).

## 2. Results and Discussion

### 2.1. Synthesis and Characterization of the Ligands H_2_L^Me^, H_2_L^Ph^ and H_2_L^PhNO2^

As mentioned before, *bis*thiosemicarbazones are a class of organic ligands of great interest in chemistry due to their proven versatility on coordination [1] and also for their biomedical applications [10,18] which could be improved over thiosemicarbazones by strong coordination to metal ions because of the presence of a larger number of donor atoms [8,9].

In this work, we have prepared a new family of *bis*thiosemicarbazone ligands, H_2_L^R^ (Figure 1), that feature a pyridine spacer and different R terminal substituents (R = Me, Ph, PhNO_2_). We aim to better understand the influence of the R substituent of the ligand on the final architecture of the derived helical compounds.

The ligands were prepared by the reaction between 2,6-pyridin-dicarboxaldehyde and 4-N-R-3-thiosemicarbazides (R = Me, Ph and PhNO_2_) in a 1:2 ratio, using absolute ethanol as solvent (Figure 2). The solids obtained were characterized by elemental analysis, infrared, mass spectrometry and NMR spectroscopies (Appendix A). 

### 2.2. Synthesis and Characterization of the Zinc Helicates

The neutral complexes Zn_2_(L^Me^)_2_∙2H_2_O **1**, Zn_2_(L^Ph^)_2_∙2H_2_O **2** and Zn_2_(L^PhNO2^)_2_ **3** derived from the H_2_L^R^ series were prepared by means of an electrochemical methodology (experimental Section 4.2, see for example ref. [33]). Electrochemical oxidation of a zinc plate in a conducting acetonitrile solution of the corresponding ligand afforded yellow (**1** and **2**) or orange (**3**) solids, which were characterized by different techniques, such as elemental analysis, infrared spectroscopy, X-ray diffraction, molar conductivity measurements, mass spectrometry and NMR spectroscopy. 

The characterization data are consistent with the formation of dinuclear compounds [M_2_(L^R^)_2_]. Infrared spectra of the complexes show a shift of the ν(C = N + C−N) and ν(C = S) bands, as well as modifications in their intensity compared to those of the free ligands due to the coordination. Moreover, the disappearance of some of the ν(NH) bands could be related to the deprotonation of the hydrazide groups in the ligands, thus confirming that they act in their dianionic form in these complexes. 

The zinc complexes were also characterized by ^1^H NMR using DMSO-d_6_ as solvent (Appendix A). The ^1^H NMR spectra of all these compounds reveal the coordination of the metal ion to the corresponding ligand and show certain common features:(i)The disappearance of the signal corresponding to the hydrazide NH groups (H_1_) confirms that the H_2_L^R^ ligands act in their [L^R^]^2−^ bideprotonated form in the complexes, as observed before for thiosemicarbazone helicates [28].(ii)The shift of the thioamide proton (H2) signal to fewer ppm, probably caused by the formation of intermolecular hydrogen bonds between the nitrogen and the thioamide protons of different complex units. Such shielding is more pronounced in the case of 1 due to the presence of the terminal aliphatic chain.(iii)The imine protons (H4) (H5 for 3) undergo a significant shift to more ppm because of the coordination of the imine nitrogen atoms to the metal ions.(iv)The pyridine ring protons (H3 and H5) (H3 and H7 for 3) exchange their positions when the ligand coordinates to the metal ions, as was found before [28].

#### X-ray Structures

Slow evaporation of the mother liquors from the synthesis of the Zn_2_(L^Me^)_2_∙2H_2_O **1**, Zn_2_(L^Ph^)_2_∙2H_2_O **2** and Zn_2_(L^PhNO2^)_2_ **3** complexes allowed us to achieve good-quality crystals for X-ray diffraction studies. Appendix A contains the main crystallographic data for these complexes, whereas Appendix A summarize the most relevant distances and angles. The crystal structures of the complexes [Zn_2_(L^Me^)_2_]·3H_2_O **1***, [Zn_2_(L^Ph^)_2_]·4CH_3_CN **2*** and [Zn_2_(L^PhNO2^)_2_] **3*** are shown in Figure 1, Figure 2 and Figure 3.

All three compounds are dinuclear zinc complexes with a similar helicate-type structure. For that reason, a joint discussion of the three compounds will be made, highlighting in each case similarities and differences.

In all of them, two strands of the dianionic ligand are helically wrapped around two Zn(II) ions, fitting the requirements to be considered helicates. However, the microstructure of the helicates, e.g., the internal mode of coordination of the two Zn(II) ions, is different in **1***/**2*** and **3***, being considered coordination isomers.

The substituted methyl and phenyl derivatives are [5+5] dihelicates: the two zinc ions are [SNNNS] pentacoordinate with a square-based pyramidal geometry [τ = 0.08 for Zn1 and Zn1^i^ in [Zn_2_(L^Me^)_2_]·3H_2_O **1*** and τ = 0.105 for Zn1 and 0.104 for Zn2 in [Zn_2_(L^Ph^)_2_]·4CH_3_CN **2*** [34] via a pyridine nitrogen atom, both imine nitrogen atoms and the two thioamide sulfurs. Each ligand strand uses one imine nitrogen atom and one thioamide sulfur to bond to each zinc atom. A rotation around the C-C bond adjacent to the pyridine ring allows the pyridine nitrogen atom of each ligand strand to be coordinated to different Zn(II) ions. An additional C-C rotation allows the remaining imine nitrogen and thioamide sulfur atoms to be coordinated to the second zinc ion, thus generating a *bis*helicoidal [5+5] structure.

In contrast, the substituted nitrophenyl derivative [Zn_2_(L^PhNO2^)_2_] **3*** is a [4+4] dihelicate. In this case, the two zinc ions are [SNNS] tetracoordinated with a distorted tetrahedral geometry via both imine nitrogen atoms and the two thioamide sulfurs atoms. Each ligand strand uses one imine nitrogen atom and one thioamide sulfur to bond to each zinc atom in a similar manner to that discussed for the substituted phenyl derivative **2***. However, in this case the rotations leading to the *bis*helicoidal architecture put the two pyridinic nitrogen atoms too far apart to bond to the two metal ions, remaining uncoordinated.

The different coordination isomers that have been found in the solid state for the *bis*thiosemicarbazone zinc dihelicates with pyridine spacer have been analyzed in the literature. Thus, structures have been found of the types [6+6] [29,31], [6+4] [28,29], [5+5] [32] and [4+4] [30]. In some cases, two coordination isomers have been isolated for the same ligand, indicating that the energy differences between them should be small. The reported and the herein-described results seems to corroborate that the introduction of different substituents in the 4-N terminal position of the *bis*thiosemicarbazones is not a determining factor for obtaining a particular coordination isomer in the case of this type of zinc dihelicate.

### 2.3. Toxicity Assays

As a previous step to the biological studies, toxicity assays with erythrocytes were carried out with the helical complexes **1**, **2** and **3**. We aimed to find out if the terminal substituent R influences in the toxicity of the compounds, therefore establishing a structure–toxicity relationship.

The low solubility of the helicates in water made the addition of a small amount of DMSO necessary. For that reason, the effect of the addition of this solvent in the culture medium with respect to cell survival was previously tested as a control. The results obtained indicate that the percentage of live cells, although decreasing to 70–80% at 48 h due to the use of DMSO, are still significant for toxicity studies.

Maintenance of erythrocytes in culture conditions for 24–48 h (Figure 4, Figure 5 and Figure 6) does not produce significant changes, a complete survival being observed in the case of **1**, whereas the survival rate lies in the interval of 55–80% in **2** and 65–80% in **3**. When the cultures are maintained for 96 h, the three complexes diminish the survival rate but is still relevant. From these data we could stay that helicate **1** is the less toxic drug for a healthy cell line, even at high concentrations and at long assay times.

### 2.4. Interaction with Proteins and Oligonucleotides 

Interaction studies of the zinc(II) helicates **1**–**3** with proteins and oligonucleotides were performed by ESI MS-TOF [35]. The details of these experiments are given in the Section 4.2. Helicates **2** and **3** showed medium solubility in DMSO at the concentration required to prepare the different incubations, the use of ultrasound and temperature being necessary to prepare the stock solutions. The different incubations with proteins and oligonucleotides (especially at 1:5 and 1:10 ratios) showed slight precipitation of the complexes, requiring centrifugation of the samples prior to injection into the mass instrument. In the case of helicate **1**, no precipitation was observed during the preparation of the stock solution.

The three tested helixes showed similar results, so as a representative example in Figure 7 we show the mass spectra obtained during the titration of albumin, myoglobin, cytochrome C and transferrin proteins, and double-stranded oligonucleotides (DS) stock solutions with helicate **1** (see Appendix A for more details).

The results obtained reveal that none of the compounds analyzed show a significant covalent interaction with the tested proteins and oligonucleotides under the conditions used. No peaks corresponding to the addition of the ligand/helicate with the biomolecules or significant variations in relative intensities were detected. This may be attributed to three factors:-No covalent interaction and thus no adduct formation between the helicate and the protein entity and/or DS.-Insoluble species or non-ionic species are formed and, therefore, cannot be observed in the MS spectrum.-There is such a weak interaction that, in the case of a binding of the helicate or part of it, is broken upon application of the ionization potential of the MS.

The experiments performed with albumin were the only case in which a change in the intensity of the target peaks was observed, so we could consider that some type of notable non-covalent interaction took place. 

As a general conclusion, we must say that the lack of interaction of the helicates with the tested biomolecules confirms the stability of these helical species in biological media. Because of this, a more specific mechanism of action could be expected from their interaction with different tumor cell lines.

### 2.5. Cytotoxicity Studies

The in vitro cytotoxic activity of the zinc helicates **1**–**3** was tested against the human cell lines MCF-7 (human breast carcinoma), A2780 (human ovarian carcinoma cells), NCI-H460 (human lung carcinoma cells) and MRC-5 (normal human lung fibroblasts), comparing the resulting IC_50_ values with those of cisplatin. 

IC_50_ values of the helicates at 48 h were obtained by an MTT (3-(4,5-dimethyl-thiazol-2-yl)-2,5-diphenyl tetrazolium bromide) colorimetric assay. All helicates are insoluble in water so a mixture of DMSO/water was used to perform the cytotoxic studies using PBS (1 mM) as buffer. The results are summarized in Table 1 and represented in Figure 8.

As can be seen from the data, the methyl helicate **1** is the most active in the three tumor lines tested, improving the results of cisplatin for the three cancer cell lines. The phenyl helicate **2** shows better IC_50_ values than cisplatin for the MCF-7 line, a similar value for the A2780 line and much lower activity for the NCI-460 line. However, nitrophenyl helicate **3** shows low activity in the three lines. Moreover, to check that these helicates do not induce a high cytotoxic activity in normal cells, IC_50_ values in normal lung fibroblasts were also studied, resulting in a minimal reduction in the normal cell viability that suggests certain selectivity against cancer cells over the healthy ones.

The cytotoxicity of the helicates **1**–**3** against three different cancer cell lines were assessed in 48 h experiments. Thus, the 4-N-methyl-substituted helicate **1** exhibited the best values of IC_50_ in the three cancer cell lines, indicating that a small alkyl group favors the cytotoxic activity whereas big aromatic substituents clearly worsen the results. A similar structural correlation was found before in the IC_50_ data obtained for phosphino-thiosemicarbazone Au(I) complexes [36]. However, the influence of the ionic/neutral nature of the complexes could not be assessed in this case because of the non-helical nature of the *bis*thiosemicarbazone complexes containing the neutral ligands with anions acting as ligands or counterions [37].

## 4. Materials and Methods

2,6-pyridin-dimethanol, manganese(IV) oxide, 4-methyl-3-thiosemicarbazide, 4-phenyl-3-thiosemicarbazide, 4-(4-nitrophenyl)-3-thiosemicarbazide, metal plates and solvents were purchased from commercial sources and were used without any purification. Melting points were determined using a BUCHI 560 instrument. Elemental analysis of compounds (C, H, N and S) was performed with a CARLO ERBA EA 1108 Analyzer. Negative electrospray ionization (ESI^−^) mass data were registered using a Bruker Microtof mass spectrometer, while MALDI-TOF mass data were registered by a Bruker AUTOFLEX using DCTB as matrix. A Varian Inova 400 spectrometer was employed to record the ^1^H NMR spectra operating at room temperature using DMSO-d_6_ as deuterated solvent. The ^13^C NMR experiments in deuterated DMSO were performed on a Bruker AMX-500. Chemical shifts were reported as *δ* (in ppm). Infrared spectra were recorded from 400 to 4000 cm^−1^ on a VARIAN FT-IR 670 with ATR PIKE. A Crison micro CCD 2200 conductivity meter was used to measure conductivity values from 10^−3^ M solutions in DMF at room temperature.

### 4.1. Synthesis and Characterization of Precursor PCDA and the Ligands H_2_L^Me^, H_2_L^Ph^ and H_2_L^PhNO2^

The 2,6-pyridin-dicarboxaldehyde (PDCA) spacer was synthesized as a preliminary step in the preparation of the ligand following the reported procedure [38].

H_2_L^Me^: 6-pyridin-dicarboxaldehyde (0.44 g, 3.2 mmol) was reacted with 4-methyl-3-thiosemicarbazide (0.68 g, 6.4 mmol) in 200 mL absolute ethanol, using one drop of HCl(ac) as catalyst. The solution formed was kept at reflux for 4 h. The azeotropic ethanol/water mixture was periodically purged using a Dean–Stark manifold. After 4 h, the solution was cooled, resulting in the formation of a yellow precipitate which was filtered under vacuum, dried and characterized. Yield: 0.95 g (95%); m.p.: 235 °C; elemental analysis: % theoretical (C_10_H_12_N_7_S_2_) C 42.7; N 31.7; H 4.9; S 20.7; experimental C 43.1; N 31.1; H 4.5; S 20.3; IR (cm^−1^) ν: 3304 m, 3160 m (N-H); 1538 s, 1512 s, 1455 s (C = N + C-N); 1159 s, 812 w (C = S); 1041 m (N-N); ESI- (*m*/*z*): 308.3 [H_2_L^Me^-H]^−^; ^1^H-NMR (DMSO-d_6_, δ (m, nH, Hx, J)): 11.76 (s, 2H, H_1_), 8.71 (q, 2H, H_2_, J = 4.4 Hz), 8.25 (d, 2H, H_3_, J = 7.8 Hz); 8.06 (s, 2H, H_4_); 7.90 (t, 1H, H_5_, J = 7.8 Hz); 3.04 (d, 6H, H_6_, J = 4.4 Hz); ^13^C-NMR (DMSO-d_6_, ppm): 178.3 (C = S), 153.3 (C = N), 141.3 (C_ar_), 137.5 (CH_ar_), 120.6 (CH_ar_), 31.26 (CH_3_).

H_2_L^Ph^: 2,6-pyridin-dicarboxaldehyde (0.31 g, 2.3 mmol) was reacted with 4-phenyl-3-thiosemicarbazide (0.77 g, 4.6 mmol) in 200 mL absolute ethanol, using p-toluensulfonic acid as catalyst. The reaction mixture was stirred under reflux with a Dean–Stark trap for 4 h. After 4 h, the resulting solution was cooled and the yellow solid obtained was filtered off, dried and characterized. Yield: 0.90 g (90%); m.p.: 220 °C; elemental analysis: % theoretical (C_21_H_19_N_7_S_2_) C 58.2; N 22.6; H 4.4; S 14.8; experimental C 58.3; N 22.3; H 4.1; S 14.2; IR (cm^−1^) ν: 3244 m, 3121 m (N-H); 1536 s, 1515 s, 1446 s (C = N + C-N); 1183 s, 807 w (C = S); 1078 m (N-N); ESI- (*m*/*z*): 432.7 [H_2_L^Ph^-H]^−^; ^1^H-NMR (DMSO-d_6_, δ (m, nH, Hx, J)): 12.11 (s, 2H, H_1_), 10.30 (s, 2H, H_2_); 8.46 (d, 2H, H_3_, J = 7.8 Hz), 8.19 (s, 2H, H_4_); 7.91 (t, 1H, H_5_, J = 7.8 Hz); 7.56 (d, 4H, H_6_, J = 7.6 Hz); 7.40 (t, 4H, H_7_, J_1_ = 8.0 Hz, J_2_ = 7.6 Hz); 7.24 (t, 2H, H_8_, J = 8.0 Hz); ^13^C-NMR (DMSO-d_6_, ppm): 177.1 (C = S), 153.7 (C = N), 143.1 (C_ar_), 139.6 (Car), 137.7 (CH_ar_), 128.8 (CH_ar_), 126.8 (CH_ar_), 126.3 (CH_ar_), 121.8 (CH_ar_).

H_2_L^PhNO2^: 2,6-pyridin-dicarboxaldehyde (0.25 g, 1.9 mmol) was reacted with 4-(4-nitrophenyl)-3-thiosemicarbazide (0.79 g, 3.70 mmol) in 200 mL absolute ethanol, using p-toluensulfonic acid as catalyst. The reaction mixture was stirred under reflux with a Dean–Stark trap for 4 h. After 4 h, the resulting solution was cooled and the orange solid obtained was filtered off, dried and characterized. Yield: 0.76 g (79%); m.p.: 260 °C; elemental analysis: % theoretical (C_21_H_17_N_9_O_4_S) C 48.2; N 24.1; H 3.3; S 12.2; experimental C 47.9; N 24.1; H 3.2; S 11.8; IR (cm^−1^) ν: 3296 m, 3126 m (N-H); 1539 f, 1504 f, 1480 f (C = N + C-N); 1193 f, 845 m (C = S); 1088 m (N-N); ESI- (*m*/*z*): 522.1 [H_2_L^PhNO2^-H]^−^; ^1^H-NMR (DMSO-d_6_, δ (m, nH, Hx, J)): 12.43 (s, 2H, H_1_), 10.57 (s, 2H, H_2_), 8.46 (d, 2H, H_3_, J = 7.8 Hz); 8.30–8.20 (m, 6H, H_4_ + H_5_); 8.10–7.90 (m, 1H, H_6_ + H_7_); ^13^C-NMR (DMSO-d_6_, ppm): 176.4 (C = S), 153.3(C = N), 145.7 (C_ar_), 144.2 (C_ar_), 144.0 (C_ar_), 137.5 (CH_ar_), 125.3 (CH_ar_), 124.2 (CH_ar_), 121.8 (CH_ar_).

### 4.2. Synthesis and Characterization of the Zinc(II) Helicates

The *bis*thiosemicarbazone zinc(II) neutral helicates were prepared by an electrochemical methodology. Since these ligands are moderately soluble in acetonitrile, the ligands were dissolved in this solvent, applying a slight heating before starting the synthesis. A current intensity of 5 mA and potential values between 7 and 12 V were used.

Zn_2_(L^Me^)_2_·2H_2_O **1**: To a solution of H_2_L^Me^ (0.05 g) in acetonitrile (80 mL) a small amount of tetraethylammonium perchlorate was added as a conductive electrolyte. This mixture was electrolyzed at 5 mA and 8.5 V at room temperature for 1 h 44 min. The electrochemical cell can be schematized as Pt(-)|H_2_L^Me^ + CH_3_CN|Zn(+). The yellow solid obtained was filtered, washed with ethyl ether and dried under vacuum. Yield: 0.056 g (88%). Caution! Perchlorate salts are potentially explosive and should be handled with care. Electronic efficiency (Ef = 0.4 mol/F^−1^). Yellow crystals suitable for X-ray diffraction studies of Zn_2_(L^Me^)_2_·3H_2_O **1*** were obtained from the mother liquors of the synthesis. M.p.: >300 °C; elemental analysis: % theoretical (Zn_2_C_22_H_34_N_14_S_4_O_2_) C 33.6; N 25.0; H 4.4; S 16.3; experimental C 34.9; N 26.3; H 4.6; S 16.7; IR (cm^−1^) ν: 3477 a, 3324 f (O-H)/(N-H); 1536 s, 1531 s, 1455 s (C = N + C-N); 1163 s, 808 w (C = S); 1039 m (N-N); MALDI+ (*m*/*z*): 373.0 [ZnL^Me^ + H]^+^, 745.0 [Zn_2_(L^Me^)_2_ + H]^+^; ^1^H-NMR (DMSO-d_6_, δ (m, nH, Hx, J)): 8.26 (s, 2H, H_4_), 7.83 (t, 1H, H_5_, J = 7.7 Hz), 7.48 (d, 2H, H_3_, J = 7.7 Hz), 7.22 (sa, 1H, H_2_), 2.75 (d, 6H, H_6_, J = 4.2 Hz).

Zn_2_(L^Ph^)_2_·2H_2_O **2**: To a solution of H_2_L^Ph^ (0.05 g) in acetonitrile (80 mL) a small amount of tetraethylammonium perchlorate was added as a conductive electrolyte. This mixture was electrolyzed at 5 mA and 10 V at room temperature for 1h 14 min. The electrochemical cell can be schematized as Pt(-)|H_2_L^Ph^ + CH_3_CN|Zn(+). The yellow solid obtained was filtered, washed with ethyl ether and dried under vacuum. Yield: 0.054 g (91%). Electronic efficiency (Ef = 0.4 mol/F^−1^). Yellow crystals suitable for X-ray diffraction studies of Zn_2_(L^Ph^)_2_·4CH_3_CN **2*** were obtained from the mother liquors of the synthesis. M.p.: >300 °C; elemental analysis: % theoretical (Zn_2_C_42_H_38_N_14_O_2_S_4_) C 49.0; N 19.0; H 3.7; S 12.5; experimental C 46.1; N 18.6; H 3.5; S 12.3; IR (cm^−1^) ν: 3397 w, 3289 w (O-H)/(N-H); 1523 s, 1495 s, 1453 s (C = N + C-N); 1185 m, 799 w (C = S); 1074 m (N-N); ESI+ (*m*/*z*): 493.0 [ZnL^Ph^ + H]^+^, 930.2 [Zn(L^Ph^)_2_ + 3H]^+^, 991.06 [Zn_2_(L^Ph^)_2_ + H]^+^; ^1^H-NMR (DMSO-d_6_, δ (m, nH, Hx, J)): 9.34 (s, 2H, H_2_), 8.56 (s, 2H, H_4_), 7.99 (d, 1H, H_5_, J = 7.8 Hz), 7.84 (d, 4H, H_6_, J = 7.8 Hz), 7.70 (d, 2H, H_3_, J = 7.8 Hz), 7.30 (t_a_, 4H, H_7_, J_1_ = 7.8 Hz, J_2_ = 8.0 Hz), 7.01 (t_a_, 2H, H_8_, J_1_ = 7.4 Hz, J_2_ = 6.9 Hz).

Zn_2_(L^PhNO2^)_2_ **3**: To a solution of H_2_L^PhNO2^ (0.05 g) in acetonitrile (80 mL) a small amount of tetraethylammonium perchlorate was added as a conductive electrolyte. This mixture was electrolyzed at 5 mA and 9.4 V at room temperature for 1h 2 min. The electrochemical cell can be schematized as Pt(-)|H_2_L^PhNO2^ + CH_3_CN|Zn(+). The orange solid obtained was filtered, washed with ethyl ether and dried under vacuum. Yield: 0.041 g (73%). Electronic efficiency (Ef = 0.5 mol/F^−1^). Orange crystals suitable for X-ray diffraction studies of Zn_2_(L^PhNO2^)_2_ **3*** were obtained from the mother liquors of the synthesis. M.p.: >300 °C; elemental analysis: % theoretical (Zn_2_C_42_H_30_N_18_O_8_S_4_) C 43.0; N 21.5; H 2.6; S 10.9; experimental C 44.5; N 21.5; H 2.6; S 10.9; IR (cm^−1^) ν: 3297 m (N-H); 1540 m, 1505 m, 1445 s (C = N + C-N); 1194 m, 845 w (C = S); 1088 m (N-N); ESI+ (*m*/*z*): 586.0 [ZnL^PhNO2^ + H]^+^, 648.9 [Zn_2_L^PhNO2^-H]^+^; ^1^H-NMR (DMSO-d_6_, δ (m, nH, Hx, J)): 10.11 (s, 2H, H_2_), 8.80 (s, 2H, H_5_), 8.17 (d, 1H, H_4_, J = 7.7 Hz), 8.12 (t, 1H, H_7_, J = 7.7 Hz), 8.05 (d, 4H, H_6_, J = 9.3 Hz), 7.82 (d, 2H, H_3_, J = 7.7 Hz).

### 4.3. X-ray Crystallography

Suitable crystals were collected from the mother liquors of the synthesis of the three helicates. X-ray diffraction data were collected on a BRUKER APPEX-II diffractometer equipped with a CCD detector using a MoK (α) graphite monochromator (λ = 0.71073 Å) under 100 K. Data reductions were performed on APPEX2 (BRUKER AXS, 2005). In all cases, an absorption correction (SADABS) was applied to the measured reflections. All structures were solved using *SIR97* (Giacovazzo et al., 1997), and refined using *SHELXL97* (Sheldrick, 2008). Hydrogen atoms were placed in calculated positions with fixed isotropic thermal parameters and included in the structure factor calculations in the final stage of full-matrix least-squares refinement. The figures included were prepared using the program Mercury. CCDC no. 2231715, 2231713 and 2231722 contain the supplementary crystallographic data for the helicates Zn_2_(L^Me^)_2_·3H_2_O **1***, Zn_2_(L^Ph^)_2_·4CH_3_CN **2*** and Zn_2_(L^PhNO2^)_2_ **3***.

### 4.4. Toxicity Assays

The toxicity assays of the helicates were examined using a Leica optical microscope. The helicates were dissolved in DMSO as stock solution.

#### 4.4.1. Extraction and Culture 

Blood collection was performed by venous puncture in a peripheral line using K2EDTA tubes to avoid clotting. Erythrocytes were immediately isolated by centrifugation using the density gradient Ficoll Paque Plus, followed by washing off the cells with RPMI 1640 serum-free medium. The isolated erythrocytes were cultured in RPMI 1640 medium supplemented with glutamine (1 mM) and 10% of FBS (fetal bovine serum). Cells were distributed in culture flasks with a density of 5 × 106 cells/mL and a temperature of 37 °C during the entire process.

#### 4.4.2. Cell Viability Assessment

Different amounts of the metal compounds were added to each culture flask; the concentrations used were 10, 100, 500 and 1000 nM, by performing 283 erythrocyte counts at different times to assess both the effect of the compound and of the incubation time on cell viability. A cell counting chamber (Neubauer) and Trypan Blue as vital staining agent were used to count the erythrocytes, and viable erythrocytes were defined as intact discocytes.

#### 4.4.3. Toxicity Studies

All experiments were carried out in triplicate and under the same culture conditions (density and temperature). In each test, in addition to the specific cultures of the compound analyzed, parallel cultures of control erythrocytes were carried out in RPMI 1640 medium supplemented with glutamine (1 mM) and 10% FBS, as well as cultures to which DMSO (100 μM) was added in a similar amount as those cultures with treatment, being indispensable to solubilize the metal compounds studied.

#### 4.4.4. Statistical Analysis

All the results were analyzed using Sigma plot software. The one-way variance analysis (ANOVA) was used to determine the existence of significative differences between different groups. A value of *p* < 0.01 (n ≥ 3) was considered significative.

### 4.5. Interactions with Proteins and Oligonucleotides Studies

The interaction studies of the zinc helicates with proteins and oligonucleotides were performed by ESI MS-TOF following the procedure described below.

The lyophilized proteins used in the assays were bought from Sigma-Aldrich: human serum albumin, transferrin, myoglobin and C cytochrome. The stock solutions of the proteins were prepared in water while the solutions of the compounds to be studied were prepared in DMSO. 

Different samples of the compounds in a ratio protein:helicate 1:1, 1:5 and 1:10 were incubated in presence of NH_4_HCO_3_ buffer (25 mM, pH 7.0), at 37 °C for 24 h. In all cases, the resulting amount of DMSO in the sample was less than 2% to avoid interference in the mass spectra signal. Furthermore, the same amount of DMSO was added in the prepared blanks of each protein.

The simple-strands of oligonucleotides were acquired from Eurofins MWG Synthesis GmbH (OP1, 5′-CACTTCCGCT-3′ y OP2, 5′-AGCGGAAGTG-3′). The double strand (DS) was synthetized mixing equimolar amounts of both simple strands, at 70 °C for 2 h, and kept at room temperature overnight. Different ratios of DS:helicate (1:1, 1:5 and 1:10) were incubated at 37 °C for 24 h, in presence of NH_4_HCO_3_ buffer (25 mM, pH 7.0).

The samples were analyzed by ESI MS-TOF, coupled to a HPLC pump, with a mobile phase of pH 7.0. The analytical conditions of TOF were 4500 V and 100 °C.

### 4.6. Citotoxicity Studies

Cytotoxicity studies of the zinc helicates were carried out in different cell lines (NCI-H460 human lung carcinoma cells, MCF-7 human breast carcinoma, A2780 human ovarian carcinoma cells and MRC-5, normal human lung fibroblasts). These cells were cultured with RPMI 1640 growth medium supplemented with 10% FBS (fetal bovine serum) in an atmosphere of 95% air and 5% CO_2_, at a temperature of 37 °C. All cell lines were provided by the USEF service (University of Santiago de Compostela, USC).

The inhibition of cell growth induced by the compounds was evaluated by the MTT assay, where a system based on tetrazolium salts of MTT (3-[4,5-dimethylthiazol-2-yl]-2,-5-diphenyltretrazolium bromide) is used for its ability to be transformed into formazan when cells are metabolically active. 

Cells were seeded in a sterile 96-well plate at a density of 4000 cells/well and incubated for 24 h in growth medium. Subsequently, the compounds dissolved in DMSO were added, maintaining the same proportion of DMSO in each well. After 96 h (at 37 °C and in an atmosphere of 5% CO_2_/95% air), 10 µL of MTT was added to each well and incubated for 4 h. MTT was prepared at a concentration of 5 mg/mL in PBS (NaCl 0.136 M, KH_2_PO_4_ 1.47 mM, NaH_2_PO_4_ 8 mM and KCl 2.68 mM).

Subsequently, 100 µL of 10% SDS in 0.01 M HCl was added and incubated for 12–14 h under the same experimental conditions. 

The ability of living cells to reduce MTT to formazan was used to detect cell viability. The absorbance was measured at 595 nm using a microplate reader (Tecan infinite M1000 PRO, Männedorf, Switzerland), considering three replicates per sample. IC_50_ values were calculated from dose-response curves using GraphPad Prism V2.01, 1996 (GraphPad Software Inc.).

## 5. Conclusions

Three new neutral zinc(II) helicates derived from *bis*thiosemicarbazone ligands Zn_2_(L^Me^)_2_∙2H_2_O **1**, Zn_2_(L^Ph^)_2_∙2H_2_O **2** and Zn_2_(L^PhNO2^)_2_ **3** were obtained by means of an electrochemical methodology and fully characterized. Their crystal structures illustrate the ability of these dihelicates to adopt different types of coordination isomerism. Some biological properties of the helicates **1**–**3** were studied with the aim of finding out whether small changes in the ligand skeleton influence their activity. The toxicity assays show that the helicates do not have a high toxicity, helicate **1** being the least toxic. Interaction studies of the three helicates with proteins and oligonucleotides by ESI MS-TOF reveal that no significant covalent interaction is stablished, proving their stability in biological media.

The in vitro cytotoxic studies of the helicates **1**–**3** against the H460, MCF-7, A2780 and MRC-5 cancer cell lines show that the methyl-substituted helicate **1** exhibit the best results, even above that of cisplatin, in all the tumor lines. These data let us to conclude that a small alkyl group as methyl promotes a low toxicity and relevant antitumor activity.

## Data Availability

Crystallographic data for **1***–**3*** were deposited into the Cambridge Crystallographic Data Centre, CCDC 2231713, 2231715 and 2231722. These data can be obtained free of charge via www.ccdc.cam.ac.uk/data_request/cif, or by emailing data_request@ccdc.cam.ac.uk, or by contacting The Cambridge Crystallographic Data Centre, 12 Union Road, Cambridge CB2 1EZ, UK; fax: +44 1223 336033.

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
