# Peer review of "Exploring the Biological Properties of Zn(II) Bisthiosemicarbazone Helicates"

_ijms, 2023, doi:10.3390/ijms24032246_

Round 1
Reviewer 1 Report
This is nice paper showing the synthesis and the cytotoxic activities of three new Zn(II) bisthiosemicarbazone helicates. X-Ray crystals were obtained and the structures are well described. However a couple of improvements are needed to make the three structure easier to follow along the paper. Standard color representation schemes using ball-sticks would help the actual structure visualization of 1-3 (blue-nitrogen, sulfur yellow, Zn purple, etc..). Also if the structures contain three molecules of water, why are they not present in the final figures?
The authors claimed that NH are not observable in the 1H NMR, but maybe the NH signals are to broad and out of range due to the Zn effect (see JACS, 1986, 108, 3298-3303). Make the feature on lines 101-103 as likely assumption.
Please, remove terms down-field and high-field as IUPAC recommends (see https://goldbook.iupac.org/terms/view/C01036), instead use more ppm or less ppm, espectively)
Author Response
This is nice paper showing the synthesis and the cytotoxic activities of three new Zn(II) bisthiosemicarbazone helicates. X-Ray crystals were obtained and the structures are well described. However a couple of improvements are needed to make the three structure easier to follow along the paper.
We thank this reviewer for the nice and pertinent comments about this paper. We have revised the paper according to the comments of this referee and we have also tried to address this reviewer concerns. These changes have undoubtedly improved the quality of this work.
Comment: Standard color representation schemes using ball-sticks would help the actual structure visualization of 1-3 (blue-nitrogen, sulfur yellow, Zn purple, etc..).
We have made the suggested change to the schemes of the three helicate structures in order to facilitate visualization.
Comment: Also if the structures contain three molecules of water, why are they not present in the final figures?
Structures 1 and 2 do indeed feature solvent molecules. However, they were removed from the representation for clarity purposes. In this sense, the sentences " Solvent molecules were omitted for clarity " were added to the description of Figures 1 and 2. The whole structure including solvent molecules can be accessed and downloaded from CCDC.
Comment: The authors claimed that NH are not observable in the 1H NMR, but maybe the NH signals are to broad and out of range due to the Zn effect (see JACS, 1986, 108, 3298-3303). Make the feature on lines 101-103 as likely assumption.
Herein the Referee 1 is making an important point regarding to the NH signals in 1H NMR. We have used an electrochemical methodology to obtain the zinc helicates in which the zinc metal undergoes an oxidation process, and the ligand experiences a reduction of the hidrazide NH groups. For that reason, disappearance of the NH signals in 1H NMR is always observed in this type of thiosemicarbazone neutral helicates (see for example Dalton Trans., 2005, 572-579). The latter can also be detected during the synthesis, as hydrogen releasing is observed in the cell.
Comment: Please, remove terms down-field and high-field as IUPAC recommends (see https://goldbook.iupac.org/terms/view/C01036), instead use more ppm or less ppm, respectively).
We have made the suggested change.

Reviewer 2 Report
The manuscript by Fernández-Fariña et al. described preparation of several bisthiosemicarbazone complexes of Zn(II), their characterization as well as some biological tests (antitumor activity etc).
|In general, the work is performed in due order, but the significance of content and the novelty of results are, in my opinion, more suitable for more specialized journal such as Molecules. I highly engourage submission to Molecules. For IJMS with its IF = 6+, these results are of interest for too narrow range of researchers. This is my only point of criticism.
Author Response
The manuscript by Fernández-Fariña et al. described preparation of several bisthiosemicarbazone complexes of Zn(II), their characterization as well as some biological tests (antitumor activity etc). In general, the work is performed in due order, but the significance of content and the novelty of results are, in my opinion, more suitable for more specialized journal such as Molecules. I highly engourage submission to Molecules. For IJMS with its IF = 6+, these results are of interest for too narrow range of researchers. This is my only point of criticism.
We thank this referee for the comments on this work.
